# Exploring Salinity Tolerance Mechanisms in Diverse Wheat Genotypes Using Physiological, Anatomical, Agronomic and Gene Expression Analyses

**DOI:** 10.3390/plants12183330

**Published:** 2023-09-20

**Authors:** Mohammed A. A. Hussein, Mesfer M. Alqahtani, Khairiah M. Alwutayd, Abeer S. Aloufi, Omnia Osama, Enas S. Azab, Mohamed Abdelsattar, Abdallah A. Hassanin, Salah A. Okasha

**Affiliations:** 1Department of Botany (Genetics), Faculty of Agriculture, Suez Canal University, Ismailia 41522, Egypt; abdelgawad475@yahoo.com; 2Department of Biological Sciences, Faculty of Science and Humanities, Shaqra University, Ad-Dawadimi 11911, Saudi Arabia; mmalqahtani@su.edu.sa; 3Department of Biology, College of Science, Princess Nourah bint Abdulrahman University, P.O. Box 84428, Riyadh 11671, Saudi Arabia; asaloufi@pnu.edu.sa; 4Environmental Stress Lab (ESL), Agricultural Genetic Engineering Research Institute (AGERI), Agriculture Research Center (ARC), Giza 12619, Egypt; omniaosama@gmail.com; 5Agricultural Botany Department, Faculty of Agriculture, Suez Canal University, Ismailia 41522, Egypt; enas_azab@agr.suez.edu.eg; 6Agricultural Genetic Engineering Research Institute (AGERI), Agriculture Research Center (ARC), Giza 12619, Egypt; mteima@ageri.sci.eg; 7Genetics Department, Faculty of Agriculture, Zagazig University, Zagazig 44511, Egypt; 8Department of Agronomy, Faculty of Agriculture, Suez Canal University, Ismailia 41522, Egypt

**Keywords:** abiotic stress, gene expression, proline, qPCR, salinity tolerance, wheat

## Abstract

Salinity is a widespread abiotic stress that devastatingly impacts wheat growth and restricts its productivity worldwide. The present study is aimed at elucidating biochemical, physiological, anatomical, gene expression analysis, and agronomic responses of three diverse wheat genotypes to different salinity levels. A salinity treatment of 5000 and 7000 ppm gradually reduced photosynthetic pigments, anatomical root and leaf measurements and agronomic traits of all evaluated wheat genotypes (Ismailia line, Misr 1, and Misr 3). In addition, increasing salinity levels substantially decreased all anatomical root and leaf measurements except sclerenchyma tissue upper and lower vascular bundle thickness compared with unstressed plants. However, proline content in stressed plants was stimulated by increasing salinity levels in all evaluated wheat genotypes. Moreover, Na+ ions content and antioxidant enzyme activities in stressed leaves increased the high level of salinity in all genotypes. The evaluated wheat genotypes demonstrated substantial variations in all studied characters. The Ismailia line exhibited the uppermost performance in photosynthetic pigments under both salinity levels. Additionally, the Ismailia line was superior in the activity of superoxide dismutase (SOD), catalase activity (CAT), peroxidase (POX), and polyphenol oxidase (PPO) enzymes followed by Misr 1. Moreover, the Ismailia line recorded the maximum anatomical root and leaf measurements under salinity stress, which enhanced its tolerance to salinity stress. The Ismailia line and Misr 3 presented high up-regulation of H+ATPase, NHX2 HAK, and HKT genes in the root and leaf under both salinity levels. The positive physiological, anatomical, and molecular responses of the Ismailia line under salinity stress were reflected on agronomic performance and exhibited superior values of all evaluated agronomic traits.

## 1. Introduction

Wheat (*Triticum aestivum* L.) is one of the most important cereal crops, accounting for around 20% of total calorie intake and protein in the human diet worldwide [1]. The global area used for wheat farming is 221 million hectares, which produces about 770 million metric tonnes [2]. Egypt is one of the largest importers of wheat, importing approximately 10 million tons annually [2]. Importantly, the production–consumption disparity is widening as a result of current and anticipated future population growth and climate change. Because of this, the cultivated area needs to be expanded into more marginal habitats, which are characterized by high saline levels [3,4]. In many parts of the world, soil salinity is a significant abiotic stress, and this problem is likely to worsen as a result of climate change, as rising sea levels could contribute to an increase in the salinization of coastal soils [5,6]. It significantly declines photosynthetic capacity, cell elongation, protein synthesis, and metabolic functions [7]. Salinity stress leads to nutrient imbalance by diminishing the uptake of essential elements, particularly K+, Ca2+, and Mg2+ [8]. Consequently, it hinders plant growth and development and is the primary factor in reducing wheat production worldwide [9].

Wheat plants (*Triticum aestivum* L.) are typically sensitive to salinity [10]. The genotypes that are relatively tolerant to salinity can survive in environments with high levels of sodium chloride, although this results in a significant reduction in productivity [11]. Cultivation of salt-tolerant genotypes is regarded as an efficient method for mitigating salinity conditions and achieving an acceptable production of crops [12]. Different types of wheat have various chances of adapting to salt stress and making enough grain under natural situations of salt stress. Several researchers have shown that bread wheat has a lot of genetic differences in how well it can handle salty conditions [13,14]. These experiments could be performed in controlled or simulated field conditions [15].

Various characteristics in the plants could be employed as indicators of symptoms of stressful environmental conditions [16]. The plants respond to salinity stress with characteristic modifications in their anatomy [17,18]. These adjustments have substantial impacts on the photosynthetic capacity and several processes in the plants [19,20]. Therefore, the anatomical alterations of the root and leaf are employed as indicators of symptoms of abiotic stresses [21]. Moreover, molecular analysis has become a powerful tool for the genetic identification of different crops [22]. The advantages of using DNA markers for analyzing genetic variation include the fact that the results cannot be affected by factors such as environmental circumstances, the stage of plant development, or the kind of organ [23].

Plants have developed a specific system of adjusting the movement of Na+ and K+ and their required availability for cellular functions through Na+ sequestration and enhanced K+: Na+ ratio [24]. Excessive Na+ sequestration is regulated by the vacuolar Na+/H+ antiporters localized in the membrane as they enable Na+ uptake into vacuoles, leading to a reduction in toxic Na+ in the cytosol in several plant species during salt stress [20]. The wheat vacuolar Na+/H+ antiporter gene TaNHX2 facilitates the sequestration of Na+ from the cytosol into the vacuole, thus enhancing salt tolerance significantly. High-affinity K+ transporters control the root-to-shoot transfer of Na+ by unloading Na+ from the xylem into xylem parenchyma cells, reducing Na+ and increasing K+ levels in shoots during salt stress [25]. HKTs transporters are known to be pivotal components of salt tolerance in plants. They are involved in Na+ long-distance translocation by contributing to Na+ unloading through the xylem, preventing excessive accumulation of Na+ in leaves, while the HAK transporter is a key component in K+ uptake through the root, and in K+ long-distance transport through loading and unloading in the vascular tissue [26,27,28,29,30].

In general, the ultimate goal of breeding programs is to increase under-stress productivity and quality. To achieve these objectives, approaches for detecting the salt tolerance of a large number of genotypes must be inexpensive, rapid, and simple to measure [31]. We hypothesized that the evaluated wheat genotypes possess wide variability and would respond differently to salinity conditions by modifying the efficiency of enzymatic and non-enzymatic antioxidants, anatomical root and leaf characteristics, and gene expression. Although published literature is available on assessing wheat genotypes under salinity stress, a comprehensive evaluation is required in order to explore different mechanisms for salinity tolerance employing morphological, physiological, anatomical, agronomic, and gene expression analyses. Therefore, the objective of this study was to explore the anatomical, physiological, morphological, molecular, and agronomic responses of diverse wheat genotypes to different salinity levels to explore the tolerance mechanisms against salinity stress.

## 2. Results

### 2.1. Response of Biochemical Compounds to Salinity Stress

Salinity treatments of 5000 and 7000 ppm gradually reduced the photosynthetic pigments chlorophyll *a*, *b* and carotenoids (Table 1). The intermediate level (5000 ppm) reduced chlorophyll *a*, *b* and carotenoids by 1.3%, 3.9%, and 4.2%, while the high level reduced the aforementioned characters by 4.7%, 4.6, and 5.4% in the same order compared to the unstressed plants. The genotype Misr 1 showed the minimum values in chlorophyll *a*, *b* and carotenoids. On the other hand, the Ismailia line was superior in all photosynthetic pigments, especially under the high-level treatment (7000 ppm), compared with the remaining wheat genotypes. Proline content in stressed plants was enhanced by increasing salinity stress by 192.61% and 218.87% compared to unstressed plants in all assessed wheat genotypes. The Ismailia line possessed the maximum values of overall proline levels (Table 1).

### 2.2. Response of Enzyme Activities to Salinity Stress

Substantial variations were detected among the assessed genotypes studied for enzyme activities (Figure 1). The highest value of superoxide dismutase (SOD) activity was recorded for the genotype Ismailia line (20.22), while the lowest value was recorded for Misr 3 (16.19). The high salinity level (7000 ppm) increased SOD activity in leaves of the studied wheat genotypes by 18.83% compared with the unstressed plants. The highest SOD activity was exhibited by Ismailia line plants under 7000 ppm conditions (21.86) (Figure 1A). In all assessed genotypes, rising salinity levels stimulated the catalase activity (CAT) compared to the non-saline control by 15.13%. The highest average value of CAT activity was recorded in the Ismailia line under 5000 and 7000 ppm conditions followed by Misr 1 under 7000 ppm conditions, while the lowest value was recorded in the genotype Misr 3 (Figure 1B). Combined with rising salinity, peroxidase (POX) activity adjusted non-specifically in all assessed genotypes (Figure 1C). The high salinity level (7000 ppm) substantially elevated POX activity in the Ismailia line and in Misr 1 plants. In general, increasing salinity boosted POX activity in the leaves under 7000 ppm conditions by about 5.57% compared to non-saline control plants. The highest POX activity was observed in Ismailia line plants under 7000 and 5000 ppm and in Misr 1 under 7000 ppm (Figure 1C). Polyphenol oxidase (PPO) activity changed significantly under 5000 and 7000 ppm compared with the control. Misr 1 and Ismailia line plants recorded high values (5.95 and 5.66) compared with Misr 3. The increasing salinity stimulated PPO activity in the leaves under 5000 and 7000 ppm in all genotypes by 15.89% and 15.46% compared with unstressed control plants, respectively. The greatest PPO activity was recorded in Misr 1 under 7000 and 5000 ppm conditions and Ismailia line plants under 7000 ppm conditions (Figure 1D). The highest activity of Glutathione reductase (GR) was recorded in the Misr 1 genotype (33.37), followed by the Ismailia line (28.43), while the lowest value was recorded in the Misr 3 genotype (Figure 1E).

### 2.3. Anatomical Measurements under Salinity Stress

Anatomical root measurements such as thickness of epidermis, cortex, vascular bundle and pith and number of xylem vessels of root were presented in Table 2. Plants grown under normal conditions (control) recorded the highest values of all anatomical root measurements. However, the genotype Ismailia line recorded superior values of epidermis thickness of root, cortex thickness of root, vascular bundle thickness of root, and pith thickness number of xylem vessels, which makes it more tolerant of salinity.

The leaf anatomy in Table 3 illustrates that increasing salinity levels up to 5000 ppm and 7000 ppm significantly decreased all anatomical leaf measurements except sclerenchyma tissue upper and lower vascular bundle thickness compared with unstressed plants. The Ismailia line was superior (more tolerant) in all anatomical leaf measurements: mesophyll thickness, main vascular bundle thickness, and sclerenchyma tissue upper and lower vascular bundle thickness compared with other genotypes under salinity stress.

### 2.4. Gene Expression under Salinity Stress

The results showed that, in the root, the H^+^-ATPase gene was up-regulated in the Ismailia line and in Misr 3 genotypes by 4- and 11-fold at 5000 ppm, respectively, and by 6-fold at 7000 ppm in the Misr 3 genotype (Figure 2A). On the other hand, H^+^-ATPase expression was down-regulated in Misr 1 in both 5000 ppm and 7000 ppm treatments. However, in leaves, the expression level was up-regulated by 2.8-fold at 7000 ppm in the Ismailia line and by 34- and 5-fold in Misr 3 genotype at 5000 ppm and 7000 ppm, respectively. On the other hand, H^+^-ATPase expression level was not significantly up-regulated in the Misr 1 genotype compared with the control plants (Figure 3A). The NHX2 gene in roots was up-regulated by 2- and 7-fold in the Ismailia line at 7000 ppm and 5000 ppm, respectively. In the Misr 1 genotype, its expression level was not up-regulated compared with the control plants. Interestingly, its expression level at 7000 ppm was sharply down-regulated. In Misr 3, the expression level was up-regulated by 2-fold at 5000 ppm, with no significant response compared to the control plants at 7000 ppm (Figure 2B). In leaves, the NHX2 expression level was down-regulated in both the Ismailia line and Misr 1, while being up-regulated by 5-fold at 5000 ppm in the Misr 3 genotype (Figure 3B). For the Ismailia line, expression levels of HKT1 in roots were up-regulated by 45- and 16-fold at 5000 ppm and 7000 ppm, respectively. On the other hand, Misr 3 was up-regulated by 15- and 4-fold at 5000 ppm and 7000 ppm, respectively, but in the Misr 1 genotype, the expression was up-regulated by 1.8- and 2-fold at 5000 ppm and 7000 ppm, respectively (Figure 2C). In leaves, the expression level of HKT1 was up-regulated by 2-, 2.3-, and 4-fold in the Ismailia line, Misr 1, and Misr 3 genotypes, respectively, at 5000 ppm and up-regulated by 6- and 4-fold in Ismailia and Misr 1, respectively, at 7000 ppm (Figure 3C). In roots, the expression level of HAK1 was up-regulated by 9- and 45-fold in the Ismailia line and Misr 3 genotype, respectively, and down-regulated in Misr 1 at 5000 ppm. On the other hand, the expression level was up-regulated by 3-fold in the Ismailia line while in Misr 1 and Misr 3 genotypes, the expression was almost the same as the control plants at 7000 ppm treatment (Figure 2D). In leaves, the HAK1 expression level was up-regulated by 3- and 2-fold in the Misr 1 and Misr 3 genotypes, respectively; in contrast, in the Ismailia line, the expression level was down-regulated at 5000 ppm and 7000 ppm (Figure 3D).

### 2.5. Agronomic Response to Salinity Stress

The analysis of variance displayed that there was a highly significant difference between salinity treatments as well as the assessed wheat genotypes over both growing seasons (Table 4). Under salinity treatment in all seasons, the Ismailia line displayed higher mean values for all traits than the other genotypes (Misr 1 and Misr 3). Days to 50% heading of all studied genotypes were reduced significantly under salinity conditions in the first (71.8, 71.4) and second season (73.3, 72.9) compared to the control conditions (80.0, 81.3). Misr 3 was the earliest in heading across three treatments (Table 4). The Ismailia line scored the highest plant height (91.6 cm and 89.9 cm) in both seasons compared to the other genotypes. Plant height was reduced significantly under both salinity levels compared to unstressed plants (Table 4). A number of effective tillers of Misr 1 exhibited no significant differences with the Ismailia line, while they significantly decreased in Misr 3 in both seasons. The number of effective tillers significantly decreased under 5000 and 7000 ppm compared to the control conditions (Table 4). In general, the main effect of salinity stress depicted 50 and 47.2% reduction in both seasons, respectively, compared to unstressed plants (Table 4). All wheat genotypes exhibited a significant reduction in spikelets per spike under salinity stress conditions. The Ismailia line and Misr 1 showed no significant difference in the number of spikelets per spike in both growing seasons. While the Misr 3 genotype showed a significant difference from the other two genotypes in both seasons (Table 4). The three genotypes were significantly different in number of grains per spike and spike length. The Ismailia line scored the highest spike length and number of grains per spike followed by Misr 1 and Misr 3 in both seasons. Also, the results showed that the salinity stress significantly reduced spike length and the number of grains per spike of all studied wheat genotypes compared to the normal conditions (Table 5). The heaviest seed index was assigned for Misr 3 followed by Misr 1 and the Ismailia line (Table 5). The results showed that salinity stress significantly reduced the 1000-grain weight of all studied wheat genotypes compared to the normal conditions (Table 5). The highest grain yield per plant was produced by the Ismailia line followed by the Misr 1 and Misr 3 genotypes (Table 5). Salinity stress significantly reduced grain yield per plant of all studied wheat genotypes compared to the normal conditions (Table 5). All genotypes reduced sharply under saline conditions, about 65.40% and 65.48%, compared to non-saline conditions in both seasons, respectively.

### 2.6. Na+ and K+ Contents and K+/Na+ Ratio in Wheat Leaves

The highest Na+ content was shown by Misr 3 followed by Misr 1 and the Ismailia line. While the highest K+ content, and subsequently the K+/Na+ ratio, was shown by the Ismailia line followed by Misr 1 and Misr 3 (Table 6). Salinity stress significantly increased Na+ content and, conversely, decreased K+ content and the K+/Na+ ratio in contrast to the control condition (Table 6).

## 3. Discussion

Salinity in soil or irrigation water has a devastating effect on the development and yield of wheat. Salt can be removed from the root zone by dissolving with freshwater or implementing specific agronomic practices [32]. To effectively manage saline soil and irrigation water, it is necessary to develop genetically salt-tolerant genotypes. Moreover, the development of salt-tolerant wheat genotypes has become more critical, particularly in light of abrupt climate change and continuing worldwide population growth [33,34]. Salt tolerance is a complex trait caused by a multitude of interconnected morphological, biochemical, physiological, and agronomic characteristics [5,10]. Therefore, salt-tolerant genotypes initiate protective tolerance mechanisms, such as changes in gene expression, allowing these plants to adapt their biochemical, physiological, and morphological responses to salinity stress [35]. In the present study physiological, biochemical, agronomic, and molecular responses of three diverse wheat genotypes (Ismailia line, Misr 1, and Misr 3) were assessed under different levels of salinity (5000 and 7000 ppm).

Chlorophyll and carotenoid contents are employed to explore the physiological status of plants induced by oxidative stress. Reduction in chlorophyll contents resulting from salinity stress is a general observation in many plant species as associated with the toxic effects of Na+ or Cl− on photosynthesis [4,36]. The obtained results in the present study indicated that carotenoid contents followed the same trends as that of the chlorophyll *a* and *b* contents and were considerably reduced under salinity stress. However, carotenoids are stated to be associated with salt tolerance in field crops [37]. The elevated level of chlorophyll and carotenoids in plants under salinity stress has been reported as a tolerance mechanism to maintain the leaves’ tissues against oxidative stress-caused damages [38,39,40,41]. In this perspective, the Ismailia line was superior in all photosynthetic pigments and was more salt tolerant compared with the other wheat genotypes. Proline is a well-known osmoprotectant that accumulates in high levels in saline environments [42]. In the wheat plants introduced to salt stress, the level of endogenous proline rises to protect the plant from salt damage [43]. Consequently, the accumulation of compatible solutes as free proline is an important strategy for plants to cope with salinity stress [44]. The reactive oxygen species (ROS) are over-produced under salinity stress [45]. ROS causes carbohydrate oxidation, DNA damage, lipid peroxidation of cellular membranes, protein denaturation, inhibition of enzyme function and decreased pigment content [46].

Studies on different crops including rice, wheat, and barley, demonstrated that elevated salinity levels contributed to oxidative stress [47,48]. Thereupon, the plants respond to salt-induced oxidative stress by exhibiting multiple antioxidant enzyme activity as an important defense approach [49]. The Ismailia line and Misr 1 showed increased activity of CAT, SOD, POD, PPO, and GR under salinity stress. Moreover, the maximum SOD, CAT, POD, and PPO activity was observed at 7000 ppm in all examined genotypes. From this perspective, Muthukumarasamy et al. [50], Jaleel et al. [51], Datir et al. [52] Datir et al. [52], Dionisio-Sese and Tobita [53], and Abdel Latef [54] disclosed increasing activities of SOD, CAT, POD, PPO, and GR due to salinity, particularly in salt-tolerant genotypes.

Under salinity stress, the plants undergo alterations at the cellular and tissue levels, resulting in a reduction in the size of root, stem, and leaf cells [55]. In this context, Nassar et al. [56] demonstrated that salt stress had a more noticeable impact on the phloem than on the xylem, in which the translocation of water-soluble and salt ions was critically restricted to the ground parts, and the transport of photosynthetic materials to the young roots and plant apex was decreased. Moreover, Atabayeva et al. [57] elucidated that salinity stress increased the ratio of exodermis to endodermis roots and decreased the thickness of the central cylinder. Likewise, in the present study, salinity stress substantially decreased all anatomical root and leaf measurements except sclerenchyma tissue lower and upper vascular bundle thickness compared with unstressed plants. However, the Ismailia line exhibited superior epidermis thickness of root, cortex thickness of root, number of xylem vessels, vascular bundle thickness of root, pith thickness, mesophyll thickness, main vascular bundle thickness, and sclerenchyma tissue upper and lower vascular bundle thickness compared with other wheat genotypes, which make it more salt-tolerant. Hu et al. [58] deduced that wheat leaves under 120 mM NaCl had a smaller cross-sectional area, which was attributed to a considerable reduction in the size of the midrib and large veins, as well as a drop in the number of medium and small veins. The plants display anatomical adaptation to salinity stress by altering the anatomical structure of their roots and leaves [59]. The Ismailia line was superior in all anatomical root and leaf measurements under 5000 and 7000 ppm compared with the other genotypes.

In salt-tolerant plants, an up-regulation in NHX-Na+/H+ antiporter activity fueled by the electrochemical proton gradient in the vacuole sap that H+-ATPase creates helps reduce Na+ toxicity through restricting Na+ presence to the vacuole [60]. In the present study, the Ismailia line and Misr 3 exhibited higher up-regulation of both H+ATPase and NHX2 in the root system under salinity stress. The up-regulation in shoot was only observed in the Misr 3 genotype at 5000 ppm and in the Ismailia and Misr 3 genotypes at 7000 ppm, suggesting a better root system for salt tolerance in the Ismailia line at the 5000 ppm treatment compared to the other two genotypes.

Na+ movement inside the plant is controlled and regulated by a high-affinity potassium transporter (HKT), which are transporters that function as Na+ selective uniporters playing a crucial role in regulating Na+ transport to shoots through the xylem vessels by unloading the ion from the xylem in the root into xylem parenchyma, inhibiting Na+ delivery to the shoot tissues [60]. When the Na+ accumulates in the shoot, then HKT loads Na+ into shoot phloem cells to be transferred back to roots through the downward stream [61]. At the root level, the Ismailia line showed the highest HKT up-regulation in both 5000 ppm and 7000 ppm treatments, implying less Na+ transportation to the shoot for better root salt tolerance, while in the Misr 1 genotype, the expression of the HKT gene was lower in both treatments. In the shoot, the transfer of Na+ back to the root through the phloem regulated by HKT was higher in the Misr 3 genotype at 5000 ppm and in the Ismailia line at 7000 ppm. HAK confers a distinct role in fine-tuning shoot Na+ exclusion, enabling a better adaptation to the fluctuating Na+ concentrations [62]. In the root, HAK had the highest up-regulation in the Misr 3 and Ismailia line genotype at 5000 ppm treatment, and slight up-regulation was observed only in the Ismailia line genotype at 7000 ppm treatment. In the shoot, the up-regulation of HAK in the Misr 1 and Misr 3 genotypes in both 5000 ppm and 7000 ppm treatments indicated better Na+ exclusion at shoot for both cultivars.

The assessed wheat genotypes differ considerably in leaf Na+, K+, and K+/Na+ accumulation and ratio. The accumulation of excess Na+ in leaves may influence cellular and morphological processes, as a higher Na+ content in the shoot adversatively influenced agronomic traits [63]. Several researchers reported considerable differences in wheat genotypes in leaf Na+ and K+ under salinity stress and their impact on morphological and agronomic characters [64,65,66,67]. All agronomic parameters are acceptable indicators for salinity tolerance. The assessed genotypes displayed considerable variation in their agronomic performance. The Ismailia line displayed higher agronomic performance in all studied agronomic traits than the other genotypes. The aforementioned positive physiological, anatomical, and molecular responses of the Ismailia line to salinity stress were reflected in superior agronomic performance compared to the other assessed genotypes. The agronomic traits responded differently to salinity stress. The number of effective tillers per plant, number of grains per spike, plant height, and 1000-grain weight were most affected by the increase in water salinity. Otherwise, the number of spikelets per spike, followed by spike length, had the least impact. In this context, Mansour et al. [34] and Moustafa et al. [68] pointed out that grain yield and its contributing traits are integral criteria for selecting salt-tolerant wheat genotypes. Additionally, they disclosed that yield components are determined earlier as number of spikes being more affected than traits determined later as 1000-grain weight. Furthermore, Hasan et al. [69], Nadeem et al. [70], and Khokhar et al. [71] elucidated different reductions in yield-contributing traits due to the negative influence of salinity stress.

## 4. Materials and Methods

### 4.1. Experimental Setup and Plant Material

The experiment was performed at the experimental farm of Suez Canal University, Egypt, during the winter seasons of 2019–2020 and 2020–2021 under greenhouse conditions. Three wheat genotypes, Ismailia line, Misr 1, and Misr 3, were used in this study. The pedigree, name, and origin of the studied bread wheat genotypes are shown in Appendix A. Two saline solutions were prepared using NaCl and a conductivity meter. The water from the tap was utilized as a control. Wheat seeds were planted in plastic pots with a diameter of 40 cm and a depth of 50 cm, and each pot was filled with a mixture of sand and farmyard manure in a volume ratio of 2:1. Each pot was planted with 10 seeds. The experiment layout was a factorial design with three replicates (three pots each), where the first factor was salt concentration, and the second factor was genotypes. Pots were irrigated once every 5 days (on average) with a saline solution of 5000 ppm and 7000 ppm or tap water.

### 4.2. Studied Characters

#### 4.2.1. Physiological Parameters

The photosynthesis pigments Chl *a*, Chl *b*, and carotenoids mg/g (DW) were measured in the third leaf of wheat genotypes 90 days after sowing according to [72]. Free proline content was determined in leaves (mg /100g FW) as described by Bates et al. [73]. Fresh leaf samples (100 mg) were collected to measure the activity of CAT, SOD, POX, PPO, and GR. The SOD activity was measured according to Beauchamp and Fridovich [74], CAT activity was measured using the method of [75], POX activity was measured according to Singh et al. [76], PPO activity was recorded in accordance with Fuerst et al. [77], and GR enzyme activity was assayed as described by Madhu et al. [78].

#### 4.2.2. Anatomical Measurements

Certain characteristics of transverse sections of wheat genotypes, root and flag leaf, were determined. Root and leaf grain samples were fixed in 70% formalin acetic acid (F.A.A.) solution, followed by dehydration, purging with ethyl-alcohol and xylene, infiltration, and embedding in pure paraffine wax (M.P. 56 to 58 C). Fifteen sections of the root and leaf were stained with safranin and light green using a rotary microtome. Using the image processing program Image, the sections were analyzed microscopically. Utilizing a Leica light Research Microscope model PN: DM 500/13613210 equipped with a digital camera [79]. The recorded anatomical measurements were epidermis thickness of root (µm), cortex thickness of root (µm), vascular bundle thickness of root (µm), no. of xylem vessels and pith thickness (µm), mesophyll thickness (µm), main vascular bundle thickness (µm), sclerenchyma tissue upper vascular bundle thickness (µm), and sclerenchyma tissue lower vascular bundle thickness (µm).

#### 4.2.3. RNA Extraction and qPCR Analysis

Total RNA from both roots and leaves of wheat genotypes was extracted using the TRIzol (Invitrogen, Carlsbad, CA, USA) in accordance with the manufacturer’s instructions and treated with RNase-free DNase (Promega Corporation, Madison, WI, USA). RNAs concentrations were measured with a NanoDrop™ 2000 (Thermo Fisher Scientific, Waltham, MA, USA) and 1.2 μg was used for cDNA synthesis. Synthesis of cDNA was conducted using COSMO cDNA Synthesis Kit (Willowfort, Birmingham, UK). Before quantitative PCR (qPCR) analysis, the synthesized cDNA was diluted to 1/10. Specific primers for qPCR analysis are provided in the Appendix A and Table 7. qPCRs were carried out on an Agilent Stratagene Mx3005p real-time PCR detection system using HERA SYBR^®^ Green qPCR master mix (Willowfort, Birmingham, UK) following the instructions to assess the expression level of the selected genes in all samples. In a total volume of 20 µL, expression analyses were performed with 1 µL of diluted cDNA, 10 µL of 2 SYBR Green PCR Master Mix, and 0.5 µL (10 M) of each forward and reverse primer. For amplification, the following protocol was utilized: 95 °C for 60 s, followed by 40 cycles of 95 °C for 3 s and 60 °C for 40 s. To evaluate the specificity of the qPCR products, a melting curve analysis from 65 °C to 95 °C was performed. As internal reference genes, actin-specific primers were used for normalization. The relative expression levels were determined using the 2-Ct method [80].

#### 4.2.4. Agronomic Traits

The studied agronomic characteristics were days to 50% heading, plant height (cm), number of effective tillers, number of spikelets per spike, spike length (cm), number of grains per spike, 1000-grain weight (g), grain yield per plant (g), K^+^ and Na^+^ concentrations, and K^+^/Na^+^ ratio.

### 4.3. Analysis of Variances

The analysis of variance (ANOVA) was applied to assess the variations due to salinity, genotype, and their interaction. The significance among treatment means was compared by Tukey’s HSD test at *p* ≤ 5% level of probability.

## 5. Conclusions

Substantial variations were detected among assessed genotypes for all studied physiological, anatomical, gene expression, and agronomic parameters. The positive physiological, anatomical, and molecular responses of the Ismailia line under salinity stress led to relative tolerance compared with the other evaluated wheat genotypes and, accordingly, could be employed as a novel genetic material in breeding programs for improving salt tolerance in wheat. Additionally, the studied physiological and anatomical parameters, as well as gene expression of H+ATPase, NHX2 HAK, and HKT, in the root and leaf provided efficient tools to explore tolerance to salinity in wheat.

## Figures and Tables

**Figure 1 plants-12-03330-f001:**
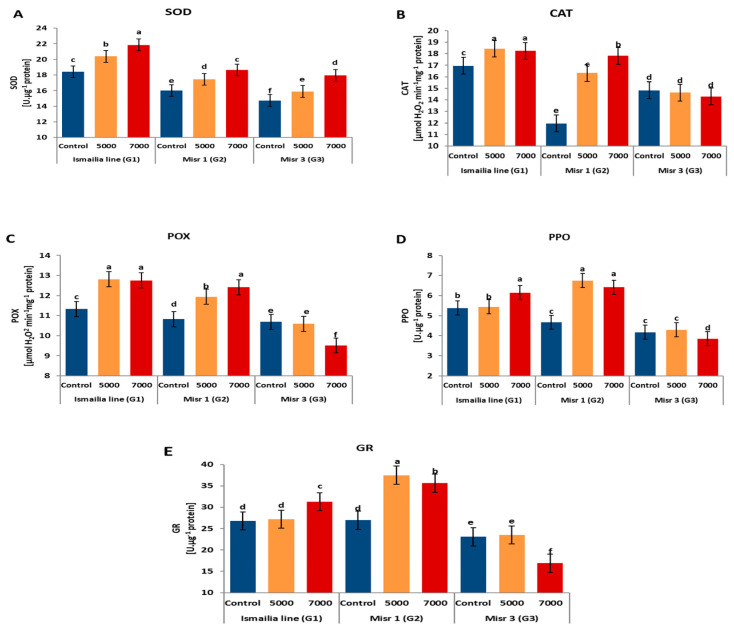
Effects of salinity treatments (control, 5000 ppm and 7000 ppm) on SOD-superoxide dismutase. (**A**) CAT: catalase, (**B**) POX: peroxidase, (**C**) PPO: Polyphenol oxidase, (**D**) GR: Glutathione reductase and (**E**) activities. Different letters differ statistically based on Tukey’s HSD test (*p* < 0.05).

**Figure 2 plants-12-03330-f002:**
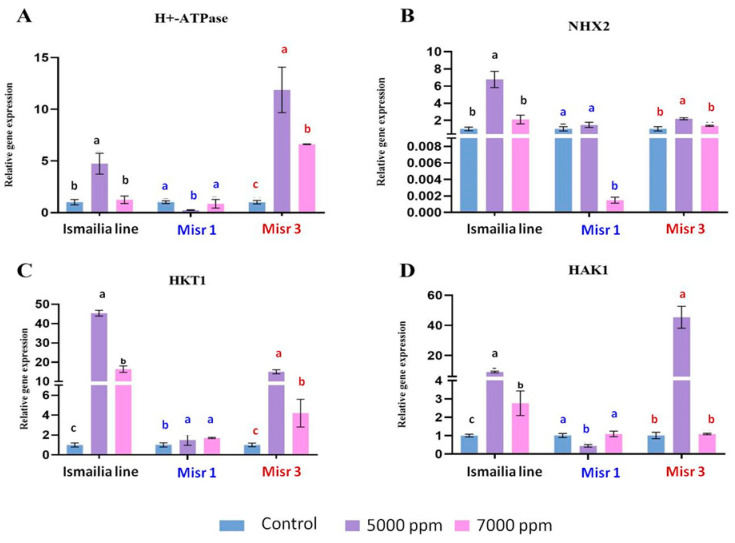
Quantitative PCR analyses of H+-ATPase, NHX2, HKT1, and HAK1 (**A**–**D**) genes from root tissues of the three wheat genotypes (Ismailia line (G1), Misr 1 (G2), and Misr 3 (G3)) under salt stress conditions (5000 ppm and 7000 ppm). Three biological replicates were performed. The error bars on the columns correspond with SD and different letters reveal significant difference at *p * <  0.05 level based on Tukey’s HSD test.

**Figure 3 plants-12-03330-f003:**
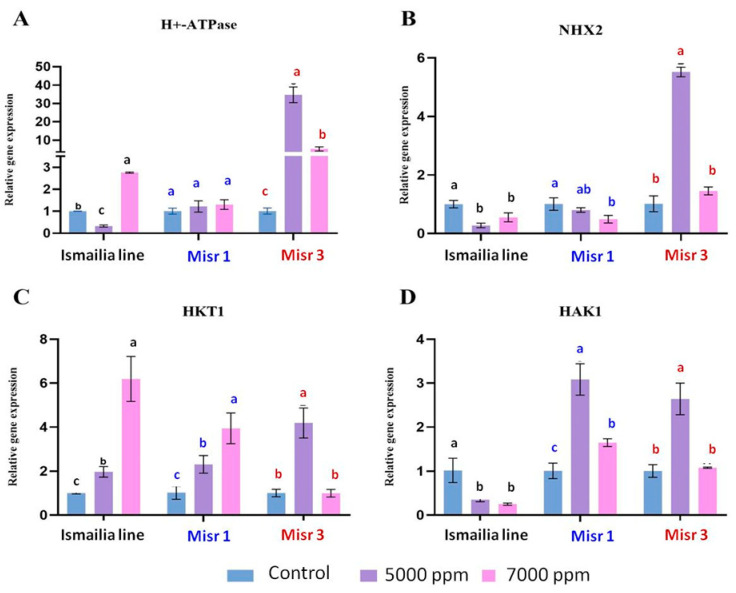
Quantitative PCR analyses of H+-ATPase, NHX2, HKT1, and HAK1 (**A**–**D**) genes from leaf tissues of the three wheat genotypes (Ismailia line (G1), Misr 1 (G2), and Misr 3 (G3)) under salt stress conditions (5000 ppm and 7000 ppm). Three biological replicates were performed. The error bars on the columns correspond with SD and different letters reveal significant difference at *p*  <  0.05 based on Tukey’s HSD test.

**Table 1 plants-12-03330-t001:** Effect of salinity levels on biochemical compound content in wheat leaves at 90 days from sowing during 2020/2021 season.

Studied Factor		Chlorophyll *a* (mg/g D.W)	Chlorophyll *b* (mg/g D.W)	Carotenoids (mg/g D.W)	Proline (mg/100 gmFW)
Wheat genotypes (G)					
Ismailia line		2.32 a	4.09 a	3.73 a	12.84 a
Misr 1		2.19 b	3.73 c	3.41 c	12.05 a
Misr 3		2.21 b	3.97 b	3.64 b	11.68 b
Salinity levels (S)					
Control		2.28 a	4.05 a	3.72 a	5.14 c
5000 ppm		2.25 a	3.88 b	3.55 b	15.04 b
7000 ppm		2.19 a	3.86 b	3.52 b	16.39 a
Interaction (G × S)					
Ismailia line	Control	2.41 a	4.21 a	3.81 a	5.11 h
	5000	2.30 ab	4.13 ab	3.66 a	15.61g
	7000	2.24 ab	3.93 bc	3.73 a	17.80 a
Misr 1	Control	2.13 b	3.95 bc	3.65 a	5.80 f
	5000	2.24 ab	3.71 d	3.21 c	14.80 e
	7000	2.20 ab	3.55 e	3.37 b	15.55 c
Misr 3	Control	2.31 ab	4.00 bc	3.70 a	4.52 i
	5000	2.20 ab	3.80 cd	3.76 c	14.72 d
	7000	2.13 b	4.10 ab	3.46 b	15.80 b
ANOVA	df	*p*-Value			
G	2	<0.001	<0.001	<0.001	<0.001
S	2	<0.001	<0.001	<0.001	<0.001
G × S	4	<0.001	<0.001	<0.001	<0.001

Means followed by different letters under the same factor are significantly different according to Tukey’s HSD test (*p* < 0.05).

**Table 2 plants-12-03330-t002:** Effect of salinity levels on some anatomical measurements (µm) of wheat roots at 90 days from sowing during 2020/2021 season.

Studied Factor		Epidermis Thickness of Root (µm)	Cortex Thickness of Root (µm)	Vascular Bundle Thickness of Root (µm)	No. of Xylem Vessels	Pith Thickness (µm)
Wheat genotypes (G)						
Ismailia line		18.7 a	222.1 a	225.2 a	15.3 a	180.9 a
Misr 1		18.8 a	92.8 b	159.4 b	14.0 b	139.7 b
Misr 3		18.1 b	217.0 a	167.0 b	11.7 c	85.7 c
Salinity levels (S)						
Control		21.4 a	238.10 a	244.23 a	14.67 a	211.43 a
5000 ppm		20.4 a	201.3 b	162.23 b	13.67 a	100.00 b
7000 ppm		13.3 b	92.41 b	145.13 c	11.67 b	94.77 c
Interaction (G × S)						
Ismailia line	Control	21.4 b	300.0 a	340.0 a	18.0 a	300.0 a
	5000	21.4 b	252.0 b	178.6 c	15.0 b	128.6 c
	7000	13.3 d	114.0 c	157.0 e	13.0 d	114.0 d
G2	Control	22.1 a	114.3 c	214.1 b	15.0 b	220.0 b
	5000	20.0 b	100.0 d	142.8 f	15.0 b	100.0 e
	7000	14.3 c	64.20 f	121.4 g	12.0 e	99.0 e
G3	Control	20.0 b	300.0 a	178.6 c	14.0 c	114.3 d
	5000	20.0 b	252.0 b	165.3 d	11.0 f	71.4 f
	7000	14.3 c	99.0 e	157.0 e	10.0 g	71.3 f
ANOVA	df	*p*-Value				
G	2	<0.001	<0.001	<0.001	<0.001	<0.001
S	2	<0.001	<0.001	<0.001	<0.001	<0.001
G × S	4	<0.001	<0.001	<0.001	<0.001	<0.001

Means followed by different letters under the same factor are significantly different according to Tukey’s HSD test (*p* < 0.05).

**Table 3 plants-12-03330-t003:** Effect of salinity levels on some anatomical measurements (µm) of wheat leaves (flag leaf) during 2020/2021 season.

Studied Factor		Mesophyll Thickness (µm)	Main Vascular Bundle Thickness (µm)	Sclerenchyma Tissue Upper Vascular Bundle Thickness (µm)	Sclerenchyma Tissue Lower Vascular Bundle Thickness (µm)
Wheat genotypes (G)					
Ismailia line		100.0 a	64.0 a	47.6 a	42.9 a
Misr 1		73.7 c	57.4 c	33.3 b	31.0 c
Misr 3		85.4 b	61.8 b	33.1 b	35.2 b
Salinity levels (S)					
Control		102.4 a	69.1 a	28.3 c	30.5 c
5000 ppm		80.9 b	59.4 b	38.1 b	35.7 b
7000 ppm		75.8 c	54.7c	47.6 a	42.8 a
Interaction (G × S)					
Ismailia line	Control	128.6 a	77.6 a	35.7 c	35.7 c
	5000	85.7 c	64.3 b	35.7 c	28.6 d
	7000	85.7 c	50.0 d	71.4 a	64.3 a
Misr 1	Control	85.7 c	65.3 b	21.4 e	28.6 d
	5000	71.4 d	50.0 d	42.9 b	35.7 c
	7000	64.1 e	57.0 c	35.7 b	28.6 d
Misr 3	Control	92.9 b	64.3 b	27.93 d	27.2 d
	5000	85.7 c	64.0 b	35.7 c	42.9 b
	7000	77.6 d	57.0 c	35.67 c	35.37 c
ANOVA	df	*p*-Value			
G	2	<0.001	<0.001	<0.001	<0.001
S	2	<0.001	<0.001	<0.001	<0.001
G × S	4	<0.001	<0.001	<0.001	<0.001

Means followed by different letters under the same factor are significantly different according to Tukey’s HSD test (*p* < 0.05).

**Table 4 plants-12-03330-t004:** Effect of salinity levels on agronomic traits of three genotypes during two seasons 2019/2020 and 2020/2021.

Studied Factor		Days to 50% Heading		Plant Height (cm)		No. of Effective Tillers		No. of Spikelets/Spike	
		2020	2021	2020	2021	2020	2021	2020	2021
Wheat genotypes (G)									
Ismailia line		78.6 a	80.3 a	91.6 a	89.9 a	2.4 a	2.8 a	16.11 a	16.2 a
Misr 1		73.2 b	74.6 b	66.1 b	64.7 b	2.2 a	2.6 a	15.3 a	15.6 a
Misr 3		71.4 c	72.7 c	64.1 b	63.0 b	1.8 b	2.1 b	13.9 b	13.9 b
Salinity level (S)									
Control		80.0 a	81.3 a	98.1 a	96.3 a	3.2 a	3.6 a	18.7 a	18.9 a
5000		71.8 b	73.3 b	64.33 b	63.0 b	1.6 b	2.0 b	13.7 b	14.1 b
7000		71.4 b	72.9 b	59.4 b	58.3 b	1.6 b	1.9 b	12.7 b	12.7 c
RD%		10.75	10.33	39.45	39.46	50.0	47.22	32.09	32.8
Interaction (G × S)									
Ismailia line	control	89.3 a	90.7 a	117.3 a	115.0 a	3.3 a	3.7 a	20.7 a	20.7 a
	5000	76.3 b	77.7 b	90.0 b	88.8 b	3.3 a	3.7 a	17.3 b	18.0 b
	7000	74.3 c	75.7 c	87.0 bc	85.3 bc	3.0 ab	3.3 a	18.3 b	18.0 b
Misr 1	control	73.3 c	75.0 c	81.7 bc	80.0 bc	1.7 bc	2.0 b	14.3 c	15.0 c
	5000	71.7 d	73.0 d	59.0 d	57.8 d	2.0 a–c	2.3 b	14.7 c	15.0 c
	7000	70.7 d	72.0 d	52.3 de	51.3 de	1.3 c	1.3 b	12.3 cd	12.3 cd
Misr 3	control	73.3 c	75.3 c	76.0 c	74.5 c	1.7 bc	2.0 b	13.3 c	13.0 cd
	5000	71.7 d	73.0 d	43.3 e	42.5 e	2.0 a–c	2.3 b	14.0 c	13.7 cd
	7000	69.3 e	70.3 e	59.0 d	57.8 d	1.3 c	1.7 b	11.0 d	11.3 d
ANOVA	df			*p*-Value					
G	2	<0.001	<0.001	<0.001	<0.001	<0.001	<0.001	<0.001	<0.001
S	2	<0.001	<0.001	<0.001	<0.001	<0.001	<0.001	<0.001	<0.001
G × S	4	<0.001	<0.001	0.034	<0.001	0.0013	<0.001	<0.001	<0.001

Means followed by different letters under the same factor are significantly different according to Tukey’s HSD test (*p* < 0.05).

**Table 5 plants-12-03330-t005:** Effect of salinity levels on agronomic traits of three genotypes in seasons 2019/2020 and 2020/2021.

Studied Factor		Spike Length (cm)		No. of Grains/Spike		1000-Grain Weight (g)		Grain Yield/Plant (g)	
		2020	2021	2020	2021	2020	2021	2020	2021
Wheat genotypes (G)									
Ismailia line		11.2 a	11.4 a	31.3 a	34.0 a	32.6 c	32.4 b	46.13 a	46.55 a
Misr 1		10.3 b	10.7 b	29.2 b	31.2 b	35.6 b	35.6 a	41.05 b	41.55 b
Misr 3		9.5 c	9.4 c	25.1 c	28.4 c	37.5 a	36.5 a	34.91 c	35.11 c
Salinity levels (S)									
Control		12.2 a	12.2 a	41.3 a	43.8 a	43.2 a	42.1 a	61.44 a	62.11 a
5000		10.16 b	10.5 b	25.9 b	28.3 b	33.9 b	33.9 b	39.38 b	39.67 b
7000		8.6 c	8.8 c	18.4 c	21.5 c	28.6 c	28.6 c	21.26 c	21.44 c
RD%		29.51	27.87	55.45	50.91	33.8	32.07	65.40	65.48
Interaction (G × S)									
Ismailia line	control	13.8 a	14.2 a	46.3 a	48.7 a	39.3 c	39.0 b	70.7 a	71.4 a
	5000	12.0 b	12.0 b	38.0 b	40.0 b	32.3 d	32.3 c	42.8 d	43.3 d
	7000	10.8b c	10.5 c	39.7 b	42.7 b	26.0 e	26.0 d	24.9 f	25.2 f
Misr 1	control	10.8b c	10.9 c	27.3 c	29.3 c	42.7 b	39.8 b	67.0 b	67.7 b
	5000	10.2 cd	10.7 c	27.3 c	29.3 c	37.0 c	37.0 bc	37.2 e	37.5 e
	7000	9.5 de	9.8 cd	23.0 d	26.3 cd	33.0 d	33.0 c	19.0 g	19.2 g
Misr 3	control	9.2 d–f	9.2 d	20.3 d	24.0 d	47.7 a	47.6 a	46.7 c	47.0 c
	5000	8.7 ef	9.3 d	22.3 d	24.3 d	32.3 d	32.3 c	38.2 e	38.7 e
	7000	8.2 f	8.0 e	12.7 e	16.3 e	26.7 e	26.6 d	19.9 g	20.1 g
ANOVA	df	*p*-Value							
G	2	<0.001	<0.001	<0.001	<0.001	<0.001	<0.001	<0.001	<0.001
S	2	<0.001	<0.001	<0.001	<0.001	<0.001	<0.001	<0.001	<0.001
G × S	4	<0.001	<0.001	<0.001	<0.001	<0.001	<0.001	<0.001	<0.001

Means followed by different letters under the same factor are significantly different according to Tukey’s HSD test (*p* < 0.05).

**Table 6 plants-12-03330-t006:** Effect of salinity levels on K+ and Na+ contents in leaves and K+/Na+ ratio of three genotypes season 2020/2021.

Studied Factor		Na+ (%)	K+ (%)	K+/Na+
Wheat genotypes (G)				
Ismailia line (G1)		8.37 c	15.83 a	2.41 a
Misr 1(G2)		9.56 b	15.29 b	1.92 b
Misr 3 (G3)		10.49 a	13.32 c	1.66 c
Salinity levels (S)				
Control		5.39 c	19.11 a	3.61 a
5000 ppm		9.44 b	15.29 b	1.64 b
7000 ppm		13.95 a	10.03 c	0.75 c
RD%		158.81	47.51	
Interaction (G × S)				
Ismailia line	Control	4.54 h	19.66 a	4.33 a
	5000	5.78 g	19.36 a	3.32 b
	7000	5.85 g	18.30 b	3.17 b
Misr 1	Control	8.31 f	16.37 c	1.97 c
	5000	9.63 e	16.20 c	1.68 d
	7000	10.39 d	13.30 d	1.28 e
Misr 3	Control	12.27 c	11.44 e	0.93 f
	5000	13.20 b	10.30 f	0.78 f
	7000	15.32 a	8.35 g	0.55 g
ANOVA	df	*p*-Value		
G	2	<0.001	<0.001	<0.001
S	2	<0.001	<0.001	<0.001
G × S	4	<0.001	<0.001	<0.001

Means followed by different letters under the same factor are significantly different according to Tukey’s HSD test (*p* < 0.05).

**Table 7 plants-12-03330-t007:** Name and sequence of primers applied for qPCR analysis.

Primer Name	Sequences (5′-3′)	Accession	Reference
TaNHX2-F	CTCCAGAACTTCGATCCTAACC	AY040246	[81]
TaNHX2-R	GCACTAAGCAATCCAGTAAACAC		
TaHKT1-F	ATGGGCCGGGTGAAAAGATT	U16709	[82]
TaHKT1-R	TCCAGAAGGGGTGAACATGC		
Ta H+-ATPase (Sub E)-F	GAGGATGCAATGAAGGAACTCC	DQ272489	[83]
Ta H+-ATPase (Sub E)-R	AAGCAGGCCCTGAACAACG		
TaHAK1-F	ACGCTTACGGGATCTGTGTG	JF495466	[82]
TaHAK1-R	GAGCCGAACACGACGTAGAA		
TaActin-F	GGAGAAGCTCGCTTACGTG	AB181991	[84]
TaActin-R	GGGCACCTGAACCTTTCTGA		

## Data Availability

The data presented in this study are available upon request from the corresponding authors.

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
