# Peer review of "Exploring Salinity Tolerance Mechanisms in Diverse Wheat Genotypes Using Physiological, Anatomical, Agronomic and Gene Expression Analyses"

_plants, 2023, doi:10.3390/plants12183330_

Round 1

Reviewer 1 Report

1- Abstract is too long and it should be revised.

2- In the abstract the full name (words) of CAT, SOD, POX, and PPO should be mentioned, after writing the full name and abbreviation form in the parenthesis, authors can write the abbreviation format.

3-Abstract should be written on the basis of Alphabetic order.

4- Introduction is too long almost one paragraph is related to Reference 2, and also Paragraph 5 just has used few references, and suddenly authors have used references 19-23, please, revise these two paragraphs.

5- In material and methods, after headline 4.2, other sub-headlines such as 4.2.1 and 4.2.2 and 4.2.3 should be Italics, for become sure, please, check the format of journal about this issue.

6- Conclusion should be extended and it can also have some information about suggestions and recommendations of authors for future researches and develop new methods for this type of experiment.

7- All scientific names both in the text and in the References should be Italics.

8- References are not the basis of journal s format. Please, correct and write all references on the basis of journal s format.

9- All articles should have DOI.

Author Response

Response to Reviewer 1 Comments

Manuscript ID: plants-2600093

Manuscript Title: Exploring Salinity Tolerance Mechanisms in Diverse Wheat Genotypes using Physiological, Anatomical, Agronomic and Gene Expression Analyses.

We are very glad and thankful for the respected Reviewer 1 for evaluating our manuscript, and providing constructive comments and valuable suggestions that have helped us further improve the quality of our manuscript. All the changes and corrections made in response to the Reviewer 1 comments were corrected as track changes in the revised manuscript. We have addressed all of your queries and improved our manuscript following your suggestions as you can see in our point-to-point responses to your comments below.

Comments and Suggestions for Authors

1- Abstract is too long and it should be revised.

Response: The abstract has been carefully revised and shortened.

2- In the abstract, the full name (words) of CAT, SOD, POX, and PPO should be mentioned, after writing the full name and abbreviation form in the parenthesis, authors can write the abbreviation format.

Response: Done accordingly.

3-Abstract should be written on the basis of Alphabetic order.

Response: The abstract has been carefully revised and the keywords have been written based on alphabetic order.

4- Introduction is too long almost one paragraph is related to Reference 2, and also Paragraph 5 just has used few references, and suddenly authors have used references 19-23, please, revise these two paragraphs.

Response: The references in the introduction have been revised and increased. The introduction has been carefully revised, shortened, and improved as requested.

5- In material and methods, after headline 4.2, other sub-headlines such as 4.2.1 and 4.2.2 and 4.2.3 should be Italics, for become sure, please, check the format of journal about this issue.

Response: Done as suggested.

6- Conclusion should be extended and it can also have some information about suggestions and recommendations of authors for future researches and develop new methods for this type of experiment.

Response: The conclusion has been revised and modified as requested.

7- All scientific names both in the text and in the References should be Italics.

Response: Response: All scientific names in the text and reference list have been revised and written in italics.

8- References are not the basis of journal s format. Please, correct and write all references on the basis of journal s format.

Response: The references have been carefully revised and formatted using the Endnote program.

9- All articles should have DOI.

Response: The DOI has been added to all references as requested.

We greatly appreciate the careful review which considerably assisted in improving the manuscript.

Reviewer 2 Report

Titled “Exploring Salinity Tolerance Mechanisms in Diverse Wheat Genotypes using Physiological, Anatomical, Agronomic and Gene Expression Analyses

Authors: Hussein et al.

Despite this huge amount of results, the paper suffers from several weaknesses that may require a major revision before a decision can be taken. Overall, in its current state, the article is merely descriptive in nature, lacking in clear hypotheses and mechanistic insight, and does not add anything new to scientific knowledge. In any case, I highly encourage the authors to carefully review point by point to clarify some issues and eventually improve the quality of the manuscript.

There some problem in this manuscript

1-    The introduction, although fluent does not present a conceptual framework able to accommodate the discussion of the results (the main one)

·        An hypotheses needs to be added at the end of the introduction and adjusted to the specific problem of the manuscript.

2-    The discussion section is very descriptive and does not contain a clear statement or idea about salt tolerance in wheat plants. It would be nice to see a hypothesis with a clear explanation and suggestion of possible mechanisms of salt tolerance.

3-    There are some sentences or information that must be cited with references

4-    The conclusion section is too long, please rewrite it focusing on the most important finding or recommendation

5- do you have any results on the osmotic adjustment or the water status of wheat in response to salt stress?

Minor editing of English language required

Author Response

Response to Reviewer 2 Comments

Manuscript ID: plants-2600093

Manuscript Title: Exploring Salinity Tolerance Mechanisms in Diverse Wheat Genotypes using Physiological, Anatomical, Agronomic and Gene Expression Analyses.

We are very glad and thankful for the respected Reviewer 2 for evaluating our manuscript, and providing constructive comments and valuable suggestions that have helped us further improve the quality of our manuscript. All the changes and corrections made in response to the Reviewer 2 comments were corrected as track changes in the revised manuscript. We have addressed all of your queries and improved our manuscript following your suggestions as you can see in our point-to-point responses to your comments below.

Comments and Suggestions for Authors

Despite this huge amount of results, the paper suffers from several weaknesses that may require a major revision before a decision can be taken. Overall, in its current state, the article is merely descriptive in nature, lacking in clear hypotheses and mechanistic insight, and does not add anything new to scientific knowledge. In any case, I highly encourage the authors to carefully review point by point to clarify some issues and eventually improve the quality of the manuscript.

There some problem in this manuscript

1-    The introduction, although fluent does not present a conceptual framework able to accommodate the discussion of the results (the main one). An hypotheses needs to be added at the end of the introduction and adjusted to the specific problem of the manuscript.

Response: The introduction has been carefully revised and improved, and the hypothesis has been added as suggested.

2-    The discussion section is very descriptive and does not contain a clear statement or idea about salt tolerance in wheat plants. It would be nice to see a hypothesis with a clear explanation and suggestion of possible mechanisms of salt tolerance.

Response: The discussion has been carefully revised and different mechanisms of salt tolerance have been discussed based on obtained results.

3-    There are some sentences or information that must be cited with references

Response: The references have been revised in the introduction and discussion and have been improved as requested.

4-    The conclusion section is too long, please rewrite it focusing on the most important finding or recommendation

Response: The conclusion has been revised, shortened, and improved as suggested.

5- do you have any results on the osmotic adjustment or the water status of wheat in response to salt stress?

Response: The physiological part of this study focused on the photosynthetic pigments, antioxidant enzyme activities (SOD, CAT, POX, and PPO) and proline content. But also in our future work, we will consider osmotic adjustment and plant water status.

Round 2

Reviewer 2 Report

Globally, the quality of the manuscript is improved; however, certain points must be resolved before a decision can be taken. The most important are:

• The absence of hypotheses or objectives at the end of the introduction; the research objectives do not describe the research carried out, but formulate the research problem, based on the state of knowledge.

• In the statistics section, the choice to use Duncan's Multiple Range Test is not appropriate for your study. In fact, Duncan's multiple range test has been denied by Professor Duncan himself. The Tukey’s method or its derivative may be recommended.

Author Response

Response to Reviewer 2 Comments (Round 2)

Manuscript ID: plants-2600093

Manuscript Title: Exploring Salinity Tolerance Mechanisms in Diverse Wheat Genotypes using Physiological, Anatomical, Agronomic and Gene Expression Analyses.

We are very glad and thankful for the respected Reviewer 2 for evaluating our manuscript, and providing constructive comments and valuable suggestions that have helped us further improve the quality of our manuscript. All the changes and corrections made in response to the Reviewer 2 comments were corrected as track changes in the revised manuscript. We have addressed all of your queries and improved our manuscript following your suggestions as you can see in our point-to-point responses to your comments below.

Comments and Suggestions for Authors

Globally, the quality of the manuscript is improved; however, certain points must be resolved before a decision can be taken. The most important are:

  • The absence of hypotheses or objectives at the end of the introduction; the research objectives do not describe the research carried out, but formulate the research problem, based on the state of knowledge.

Response: We would like to thank the Reviewer for his second revision of our manuscript and his positive assessment of the first revision. The hypotheses and  objectives have been revised and considerably improved as requested. Please see lines 100-109 in the revised version.

  • In the statistics section, the choice to use Duncan's Multiple Range Test is not appropriate for your study. In fact, Duncan's multiple range test has been denied by Professor Duncan himself. The Tukey’s method or its derivative may be recommended.

Response: The differences among the studied treatments were separated by Tukey’s HSD test instead of Duncan's multiple range test as requested.